# Reinforcement Learning with Extreme Minimum Distribution

## Abstract

Distributional Reinforcement Learning (DRL) has recently achieved remarkable performance by leveraging the distributional information of accumulated rewards. Because the data in the tail of the distribution is scarce, it is difficult to fit the tail distribution, resulting in the tail behavior of the return distribution that limits the performance of DRL. One solution to tackle the tail behavior of the return distribution is truncating the tail atoms in DRL, but this approach does not have theoretical support in previous work. We adopt the perspective of Extreme Value Theorem, which allows us to reconsider the rationality of truncating tail atoms. We further derive a conclusion: Truncating tail atoms eventually transforms the distribution generated by DRL into the extreme minimum distribution and remains stable. Building on this conclusion, we redesign the critic networks in actor-critic method. Specifically, we use two critics that output the location and scale parameters of the extreme minimum distribution to directly model the distribution of truncated DRL. Our method uses fewer networks and fewer computational resources than other truncated DRL methods and achieves more competitive experimental results than other RL methods.

## 1 Introduction

Model-free deep reinforcement learning (RL) algorithms acquire training data through agent-environment interactions, with the aim of learning an optimal policy by optimizing the cumulative discounted rewards. RL has been successfully applied to various complex domains, including game playing (Silver et al., 2016) and robotic control (Schulman et al., 2015). Current RL methods often use neural network as function approximator to solve diverse decision-making and control tasks (Mnih et al., 2013; Schulman et al., 2017). Nonetheless, numerous challenges persist in realizing the broader applicability of these methods in real-world scenarios. One of the main reasons hindering the application of these methods is the inefficient performance of algorithms. Enhancing the performance of RL for real-world deployment remains a focal area of research and attracts significant attention from the scientific community.

Distributional Reinforcement Learning (DRL) algorithms show intrinsic uncertainty during the training process of RL by estimating the entire distribution of future returns, rather than just the expected value, and achieve state-of-the-art performance in many RL benchmark tasks (Bellemare et al., 2017; Dabney et al., 2018a; Yang et al., 2019; Nguyen et al., 2020; Kuznetsov et al., 2020; Zhou et al., 2021). Currently, the mainstream distributional representations in DRL include learning discrete categorical distributions (Bellemare et al., 2017; Barth-Maron et al., 2018; Qu et al., 2019), and learning distribution quantiles Dabney et al. (2018a;b); Zhang & Yao (2019), which are referred to as Categorical Distributional Reinforcement Learning (CDRL) and Quantile Distributional Reinforcement Learning (QDRL), respectively. QDRL is commonly used in continuous control domains (Ma et al., 2020; Kuang et al., 2023) as it is simpler to implement and often yields better results than CDRL.

While DRL has shown impressive performance, there are still several aspects that can be optimized to improve the representation of the model and lead to better results (Kuang et al., 2023; Zhou et al., 2020; 2021). One aspect is regard to the tail behavior of the return distribution (Kuang et al., 2023). It is widely acknowledged that the accuracy of an estimated value is highly dependent on the quantity of data observed at a given location (Koenker, 2005). The tails of the distribution have a low

probability of being observed, which can contribute to unstable quantitative estimates in these areas. Kuznetsov et al. (2020) propose truncating the right tail of the approximated return distribution by discarding some topmost atoms, to address the common issue of overestimation (van Hasselt et al., 2016) in RL. This approach also partially addresses the tail behavior problem. It is just that the authors do not give an explanation for resolving the tail behavior problem (Kuang et al., 2023).

In statistics, the Gumbel distribution is frequently employed to model extreme events spanning diverse fields (Gumbel, 1948). We use the Gumbel distribution to justify the rationality of truncating DRL. The Extreme Value Theorem (Fisher & Tippett, 1928) provides an insight: the maximum of $i$ samples derived from a distribution with an exponential tail will asymptotically converge towards the Gumbel distribution. In DRL, we assume that certain tail atoms in our distribution can be considered as "extreme atoms". Just like extreme weather can have a significant impact on life, extreme atoms can also have a devastating effect on distribution models. Since overestimation is prevalent in RL, extreme values are often unattainable and they often cause irreversible damage to fragile network models during the update process. During the updating process, removing extreme values, including highly improbable values caused by overestimation, often enhances the stability of the model. Furthermore, since rounded random variables are sub-Gaussian (Wainwright, 2019), which have exponential tails (Garg et al., 2023), we can regard the parts that need to be removed as a set of data that conforms to the Gumbel distribution. Underpinned by this theory, we substantiate through rigorous analysis that after a sufficient number of iterations, the value distribution converges to or approximates the distribution of extreme minimum values and sustains this pattern. Extreme minimum distribution is a distribution akin to the Gumbel distribution but designed to model the smallest value in a data set.

With the previous analysis, we can re-model the value network of DRL. The revised approach no longer outputs atomic values at quantiles but instead outputs the position and scale parameters of the extreme minimum value distribution, enhancing the efficiency of model training. Based on soft actor-critic methods, we introduce the Reinforcement Learning with Extreme Minimum Distribution (EMD).

The main contributions of this work are summarized below:

- We give an intuitive explanation for solving the tail behavior problem in DRL by truncation using the Extreme Value Theorem in statistical learning.
- Through rigorous derivation, we obtain the conclusion: By continuously removing the extreme atoms in the right tail of the distribution, after a sufficient number of iterations, the distribution outputted by the value network will converge to the extreme minimum distribution and remain stable.
- We re-model the critic network to get our framework called EMD. To our knowledge, EMD is the first to use the extreme minimum distribution to model the critic network.
- According to the experimental results, our method demonstrates more competitive performance than baseline approaches in all environments within the standard OpenAI Gym (Brockman et al., 2016) benchmark suite powered by MuJoCo (Thaler, 2012).

## 2 RELATED WORK

The main component of our approach is a neural network that outputs the distribution of extreme minimum values, which can achieve the effect of using truncated DRL with fewer networks and fewer outputs. EMD is similar to the actor-critic approach in the form of its implementation. In this section, we will review previous works on DRL and actor-critic methods.

In continuous control domains, the earliest popular off-policy actor-critic method is deep deterministic policy gradient (DDPG) (Lillicrap et al., 2015), which is a variant of deterministic policy gradient algorithm (Silver et al., 2014) that learns off-policy using a $Q$-function estimator and a deterministic actor for maximizing $Q$-function. However, DDPG has two main problems.

The first one is the overestimation. The most representative for solving this problem is twin delayed deep deterministic policy gradient (TD3) algorithm (Fujimoto et al., 2018), using the clipped double $Q$-learning method. However, TD3 introduces an underestimation bias while mitigating overestimation. Furthermore, many methods are proposed to reduce estimation bias, such as SD3 (Pan

et al., 2020) and REDQ (Chen et al., 2021), which further improve the accuracy of critic network prediction.

The second problem is that the interaction between the deterministic actor network and the $Q$-function makes DDPG extremely unstable and sensitive to hyperparameter settings (Duan et al., 2016; Henderson et al., 2018). The problem limits the ability in high-dimensional tasks. To address this problem, SAC (Haarnoja et al., 2018) maximizes policy entropy while maximizing expected returns, while encouraging the use of a Gaussian policy. The introduction of the maximum entropy framework improves the exploration capability and helps SAC achieve good and stable experimental performance.

For DRL, the first algorithm is C51 (Bellemare et al., 2017), which encourages modeling the distribution of cumulative reward random variables and provides many theoretical proofs for this, becoming an important cornerstone for subsequent related research. QR-DQN (Dabney et al., 2018b) uses quantiles to represent the distribution, which not only improves the performance, but also makes the implementation of DRL more flexible. Afterwards, many works (Dabney et al., 2018a; Nguyen et al., 2020) in RL with discrete action spaces are made to improve the performance of DRL. In the continuous domain, the first application of DRL is distributed distributional deep deterministic policy gradient (D4PG) (Barth-Maron et al., 2018), which combines DDPG and C51 with distributed training, showing the potential of DRL in continuous action tasks. Distributional soft actor-critic (DSAC) (Ma et al., 2020) incorporates the idea of maximum entropy into DRL, further improving the algorithm performance. Truncated quantile critics (TQC) (Kuznetsov et al., 2020) solves the inherent overestimation problem in $Q$-learning through the truncating method, balancing estimation bias by dropping the largest atoms produced by the quantile network.

## 3 BACKGROUND

### 3.1 REINFORCEMENT LEARNING

Consider an infinite-horizon Markov decision process $(\mathcal{S}, \mathcal{A}, p, \gamma, r)$, with a set of states $\mathcal{S}$, a set of actions $\mathcal{A}$, the unknown state transition probability $p : \mathcal{S} \times \mathcal{S} \times \mathcal{A} \to [0, \infty)$ represents the probability density of the next state $s_{t+1} \in \mathcal{S}$ given the current state $s_t \in \mathcal{S}$ and action $a_t \in \mathcal{A}$. $\gamma \in [0, 1)$ is a discount factor determining the priority of short-term rewards. The environment emits a bounded reward $r : \mathcal{S} \times \mathcal{A} \to [r_{\min}, r_{\max}]$ on each transition. A policy $\pi$ maps each state $s \in S$ to a distribution over $\mathcal{A}$.

In RL, the objective is to find the optimal policy $\pi_\phi$, with parameters $\phi$, to maximize the discounted return

$$J(\phi) = \mathbb{E}_{\Gamma \sim \pi_\phi}[\sum_t \gamma^t r(s_t, a_t)]. \tag{1}$$

along a trajectory $\Gamma = (s_0, a_0, s_1, a_1, \cdots)$ obtained by executing the policy.

### 3.2 DISTRIBUTIONAL REINFORCEMENT LEARNING WITH QUANTILE REGRESSION

QR-DQN (Dabney et al., 2018b) focuses on approximating and maximizing the cumulative discounted return distribution $Z^\pi(s, a) = 1/N \sum_{n=1}^N \delta_{z_n(s,a)}$, where $\delta_{z_n(s,a)}$ denotes a Dirac at $z_n(s, a) \in \mathbb{R}$ and $z_n(s, a)$ is the $n$-th output of quantile network. The mean of the distribution is the expected value of the approximate return in the RL methods, also known as $Q$-function:

$$Q^\pi(s, a) := \mathbb{E}[Z^\pi(s, a)]. \tag{2}$$

The parameters $\theta$ of the quantile network are optimized by minimizing the averaged over the replay 1-Wasserstein distance between $Z_\theta(s, a)$ and the temporal difference target distribution $\mathcal{T}^\pi Z_\theta(s, a) := r(s, a) + \gamma Z_\theta(s', a')$, where $s' \sim p(\cdot|s, a), a' \sim \pi(\cdot|s')$. The 1-Wasserstein distance is characterized as the $L_1$ norm on inverse cumulative distribution functions (inverse CDFs) (Müller, 1997).

Let $F_V(v) = Pr(V \leq v)$ represent the CDF of distribution $V$, the inverse CDF of $V$ can define by:

$$F_V^{-1}(\omega) := inf\{v \in \mathbb{R} : w \leq F_V(v)\}. \tag{3}$$

The 1-Wasserstein metric between distributions $U$ and $V$ is given by:

$$W_1(U, V) = \int_0^1 \left| F_V^{-1}(\omega) - F_U^{-1}(\omega) \right| d\omega. \tag{4}$$

In terms of implementation, the loss founction can be performed by learning quantile locations for fractions $\tau_n = (2n-1)/2N$, $n \in [1, ..., N]$ via quantile regression. The quantile regression loss is defined as:

$$\mathcal{L}_{QR}^\tau(\theta) := \mathbb{E}_{\hat{Z} \sim \mathcal{T}^\pi Z_\theta, Z \sim Z_\theta} \left[ \rho_H^\tau \left( \hat{Z} - Z \right) \right], \tag{5}$$

where $\rho_H^\tau(u) = |\tau - \mathbb{I}(u < 0)| \mathcal{L}_H^1(u)$ and $\mathcal{L}_H^1(u)$ is Huber loss (Huber, 1964) with $\kappa = 1$ aimed to smooth the gradient at $0$:

$$\mathcal{L}_H^\kappa(u) = \begin{cases} \dfrac{1}{2} u^2, & if \ |u| \le \kappa \\[2ex] \kappa \left( |u| - \dfrac{1}{2}\kappa \right), & otherwise \end{cases} \tag{6}$$

### 3.3 EXTREME VALUE THEOREM

In statistics, the Extreme Value Theorem is also known as the Fisher–Tippett theorem, and it tells us that the maximum of i.i.d. samples from exponentially tailed distributions will asymptotically converge to the Gumbel distribution, whose probability density function ($PDF$) is $g(x) = \exp\left(-(z + e^{-z})\right)/\beta$ where $z = (x - \mu)/\beta$, $\mu$ is the location parameter and $\beta$ is the scale parameter.

**Theorem 1** *(Extreme Value Theorem) (Pranklin, 1974; Fisher & Tippett, 1928). For i.i.d. random variables $X_1, ..., X_n \sim f_X$, with exponential tails, $\lim_{n \to \infty} \max_i(X_i)$ follows the Gumbel distribution. Furthermore, $\mathcal{G}$ is max-stable, i.e. if $X_i \sim \mathcal{G}$, then $max_i(X_i) \sim \mathcal{G}$ holds.*

In other fields, the Gumbel distribution is often used to describe many extreme events, such as extreme temperatures in meteorology, extreme market volatility in finance, and so on. In RL, $\mathcal{X}$-QL (Garg et al., 2023) experimentally finds that Bellman error is closer to Gumbel distribution than Gaussian distribution, and $\mathcal{X}$-QL uses the Gumbel distribution to re-model the Bellman error and optimizes $Q$- function using a maximum likelihood approach. The Gumbel distribution is considered to have a wide range of promising applications in RL.

## 4 RE-MODELING TRUNCATED DRL USING EXTREME VALUE THEOREM

We use the Extreme Value Theorem to reconsider the practice of truncating the tail of a distribution (Kuznetsov et al., 2020). In continuous control domains in DRL, the existence of unavoidable noise in function approximations leads to the emergence of overestimated outliers, which can be effectively modeled using the Gumbel distribution. As the Extreme Value Theorem tells us, the maximum value of i.i.d. samples with an exponential tail distribution converges asymptotically to the Gumbel distribution. Therefore, we can reasonably assume that the topmost atoms of the approximated reward distribution are anomalies with large estimation bias caused by the uncertainty of the function approximators. Truncating these anomalies is a good choice when there is no good way to use them. Although this may sacrifice some information on the right tail, it can make the model updates more stable and not be impacted by extreme values.

### 4.1 EXTREME MINIMUM DISTRIBUTION

Now we further explore the application of the Extreme Value Theorem in DRL. We directly model the distribution after truncation to reduce the consumption of model representation.

**Theorem 2** *(Distribution of Truncate DRL). For a distribution of any shape, there exists a truncated number $N$ of truncations to the right tail of times, $\forall n > N$, such that the remaining distribution after $n$ truncations follows the extreme minimum distribution. In RL, truncating the maximum output of the critic network will cause the remaining portion to follow the extreme minimum distribution.*

We give the proof of Theorem 2 in Appendix A.1. There are three forms of Extreme Velue Theorem. We introduce our method in the main text using the type$-$I generalized extreme value distribution as an example. We will discuss the other two types of distributions in Appendix D. Despite sharing a similar generative process with the Gumbel distribution, the extreme minimum distribution has received little attention in the literature. Before further elaborating our approach, we provide here the theorem of the extreme minimum distribution, which is modeled after the definition of Gumbel distribution:

**Theorem 3** *(Extreme Minimum Distribution)(Fisher & Tippett, 1928). For i.i.d. random variables $X_1, \cdots, X_n \sim f_X$, with left exponential tails, $\lim_{n\to\infty} \min_i (X_i)$ follows the extreme minimum distribution $\mathcal{F}(\mu, \beta)$, which has PDF $f(x) = exp(z - e^z)/\beta$ and CDF $F(x) = 1 - \exp(-e^z)$, where $z = (x - \mu)/\beta$ with location parameter $\mu$ and scale parameter $\beta$.*

**Theorem 4** *(Min-stable)(Mood, 1963). $\mathcal{F}$ is min-stable, i.e. if $X_i \sim \mathcal{F}$, then $\min_i (X_i) \sim \mathcal{F}$ holds.*

**Theorem 5** *(Hazan & Jaakkola, 2012). For i.i.d. $\epsilon_i \sim \mathcal{F}(0, \beta)$ added to a set $\{x_1, \ldots, x_n\} \in \mathbb{R}$, $\min_i (x_i + \epsilon_i) \sim \mathcal{F}(-\beta \log \sum_i \exp(-x_i/\beta), \beta)$.*

Due to the fact that the information related to extreme minimum distribution is always passed in literature related to Extreme Velue Theorem, we provide a simple proof of Theorem 3, Theorem 4 and Theorem 5 in Appendix A.1.

Theorem 4 and Theorem 5 ensure that the minimum distribution maintains its shape and does not get converted to other distributions during the RL update process. Therefore, the distribution can be used to continuously model the truncated cumulative discounted reward.

## 4.2 Reinforcement Learning with Extreme Minimum Distribution

Starting from the perspective of truncation methods to solve the tail behavior of the return distribution in DRL, we propose a framework for modeling the value function called EMD (Extreme Minimum Distribution). In this subsection, we introduce a practical algorithm and provide a detailed explanation of the loss function.

### 4.2.1 Computation of the target distribution

In our EMD method, we train two critic networks to estimate the distribution. These networks output the location parameter and scale parameter of the extreme minimum distribution respectively. Since the network outputs the scale parameter, the sudden "caprice" of a single network can easily crash the entire model. We train two networks and limit the range of the scale parameter to minimize the impact of sudden network instability during training, which can cause the model to crash. In subsequent experiments, we will further discuss this model crash phenomenon.

To generate the target distribution, we define a target network for each critic network, following actor-critic method. Next, we consider truncating the target distribution. Truncating the distribution directly is a complex task, so in the implementation of our method, we borrow the idea of truncating the distribution from the quantile method. Specifically, we use a set of quantile locations for fractions $\tau_n = (2n-1)/2N, n \in [1, ..., N]$ and compute the inverse CDF of these points in the target distribution:

$$Z_{tar}(s, a) = F^{-1}(\tau; \mu_{tar}, \beta_{tar}) = \mu_{tar} + \beta_{tar} \ln(-\ln(1 - \tau)), \qquad (7)$$

where $\mu_{tar}$ is the location parameter and $\beta_{tar}$ is the scale parameter of the target distribution, we use these values to represent our distribution, like the atoms output by the quantile network. After this, we combine the computed quantiles of the two target distributions, and discard a small portion of the operators with the largest locations in the inverse cumulative distribution. We use $M$ to denote the number of discarded atoms. Kuznetsov et al. (2020) have demonstrated that this approach reduces the overestimation of the $Q$-function approximation, and the characteristics of extreme minimum distribution ensure that such an approach maintains the form of the extreme minimum distribution.

The overall flow of the target distribution computation is shown in Appendix B.1. For the truncated atoms obtained using the above method, we use the Bellman update equation with entropy

regularization:

$$\mathcal{T}_\pi z_n\left(s, a\right) := r + \gamma\left[z_n\left(s', a'\right) - \alpha \log \pi\left(a|s\right)\right], \tag{8}$$

to obtain temporal difference target distribution of atoms for optimizing the critic network, where $z_n(s, a)$ is the $n$-th atoms of the inverse CDF $Z(s, a)$ and $\alpha$ is the entropy temperature coefficient. At this stage, the blended target distributions of temporal differences can be regarded as

$$Y\left(s, a\right) := \frac{1}{2N - M} \sum_{n=1}^{2N-M} \delta\left(\mathcal{T}_\pi z_n\left(s, a\right)\right). \tag{9}$$

$2N - M$ represents the number of atoms remaining after discarding.

### 4.2.2 Loss functions

In the case of critic networks with output extreme minimum value distributions, we optimize by minimizing the 1-Wasserstein distance between each of $Z_i\left(s, a\right)$ $(i = 1, 2)$ represents the $i$-th extreme minimum distribution network and the temporal difference target distribution $Y(s, a)$. Same as QR-DQN (Dabney et al., 2018b), we use the approximation of quantiles in the target distribution to update the extreme minimum distributions represented by the critic network outputs $\mu_i$ and $\beta_i$. During the update, the distribution is also approximated using quantile points. For the previously mentioned fractions $\tau_n$, the loss is

$$\mathcal{L}\left(s, a; \theta_i\right) = \frac{1}{N\left(2N - M\right)} \sum_{n=1}^{N} \sum_{m=1}^{2N-M} \rho_{\tau_n}^H\left(\mathcal{T}_\pi z_m\left(s, a\right) - z_n^i\left(s, a\right)\right), i = 1, 2. \tag{10}$$

In this way, each of critic becomes dependent on all atoms of the truncated mixture of target distributions.

For policy parameters $\phi$, we utilize the maximum entropy approach, that is, we maximize the policy entropy while maximizing the expected return. By utilizing the fixed positional parameter $\mu$ and scale parameter $\beta$ of our critic network output, we can directly compute the expected discounted reward,

$$Q_i\left(s, a\right) = \int x f\left(x; \mu_i, \beta_i\right) dx = \mu_i - \lambda \beta_i, i = 1, 2, \tag{11}$$

where $\lambda = 0.5772$ is the Euler–Mascheroni constant (Sweeney, 1963), and the detailed derivation is given in Appendix A.2. We then choose the smaller $Q$ value computed by the two networks and update the policy by minimizing the loss:

$$J_\pi\left(\phi\right) = \mathbb{E}_{\mathcal{D}, \pi}\left[\alpha \log \pi_\phi\left(a|s\right) - \min_{i=1,2} Q_i\left(s, a\right)\right], \tag{12}$$

where $\mathcal{D}$ is the distribution of previously sampled states and actions, or a replay buffer, $s \sim \mathcal{D}$, $a \sim \pi_\phi\left(\cdot|s\right)$. The overall flow of our algorithm is shown in Algorithm 1 in Appendix B.2.

## 5 Experiments

The experiments in this section mainly answer the following issues: (1) Can EMD accurately fit the distribution generated by truncated DRL algorithms? (2) Can EMD achieve more competitive results among similar methods? (3) How sensitive are the hyperparameters of EMD? In addition, we provide a comparison in Appendix C.2 regarding the run time of EMD and baselines.

### 5.1 Ability to Fit Truncated DRL Approach

We answer the first question through experiments: Can EMD accurately fit the distribution generated by truncated DRL algorithms? To validate this, we add our critic network that generates the distribution of extreme minimum to TQC and use the training data generated by TQC to optimize our critic network without changing TQC. The parameters used in TQC are the same as the optimal parameters in the original paper (Kuznetsov et al., 2020). We record the location and scale parameters generated by our critic network and the 125 atoms (number of atoms output from TQC paper). Those atoms are generated by the original five critic networks of TQC at 1 million, 2 million, and 3

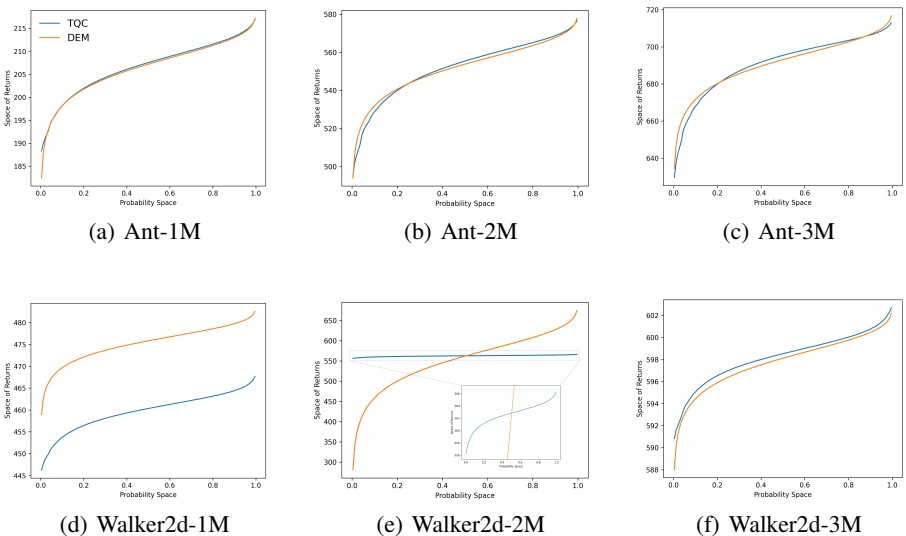

Figure 1: Ability to fit TQC.

million training steps. Then we represent them by the mean of a batch. We plot the inverse cumulative distribution function of the distributions represented by these parameters, which result is shown in Fig. 1.

In Fig. 1, we can demonstrate that the critic network of the EMD has a good fitting ability for the atoms generated by TQC. The Ant environment represents a state where the fitting is relatively stable. There is good fitting throughout the entire process. Verifying that our Theorem 2 is correct: Constantly truncating the tail atoms in the distribution will cause the distribution to approach the extreme minimum distribution and maintain. For the Walker2d environment, we need to explain that this phenomenon of fitting error during some time periods may appear in any environment. The phenomenon in Figure 1(e) appears in some seeds of any environment, and there are also seeds in the Walker2d environment that have good fitting effects throughout the process. Our single critic network may experience model collapse due to the presence of scale parameters, as shown in Figure 1(e). In this experiment, since the training data is generated only by the TQC algorithm, the collapsed network does not affect other components. So the phenomenon gradually disappears as the training progressed, and the fitting ability can be restored.

Through observing Fig. 1, we can see that the atoms generated by TQC always maintain a shape similar to the extreme minimum distribution. In addition, we can see that even though our very small value model just uses one single network to represent it, we obtain the same distribution results as the 5 large quantile networks used in TQC. Theoretically, we can use the extreme minimum distribution to replace the critic networks in TQC and achieve the same experimental results with less overhead.

## 5.2 COMPARATIVE EVALUATION

Our proposed algorithm, Extreme Minimum Distribution (EMD), builds upon the SAC by modifying the critic network to output the location parameter and scale parameter of the extreme minimum distribution. Update our critic network by reducing the 1-Wasserstein metric between distributions. We provide specific parameter settings in Appendix C.1.

To evaluate the performance of our algorithm, we compare it to related algorithms that achieve the best performance on a series of challenging continuous control tasks in the OpenAI Gym benchmark suite (Brockman et al., 2016). TQC has a similar update process to ours, but uses complex ensembling critic networks and an ensemble approach, which further increases training time. TD3, SAC and our EMD use a similar network structure, but the critic networks in TD3 and SAC have different outputs and use different loss functions for updates compared to ours. Figure 2 shows the

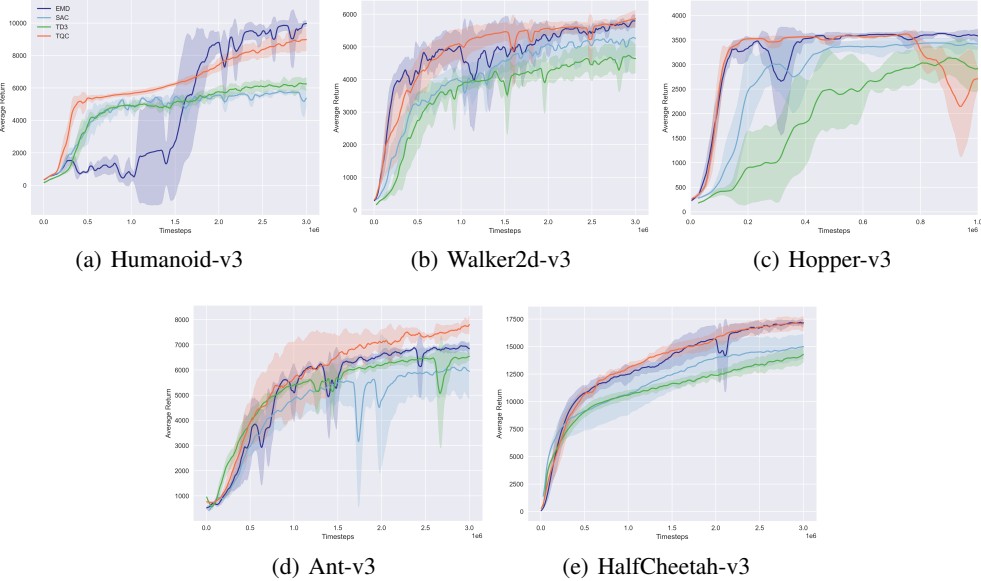

(a) Humanoid-v3          (b) Walker2d-v3          (c) Hopper-v3

(d) Ant-v3          (e) HalfCheetah-v3

Figure 2: Training curves on continuous control benchmarks.

total average return of evaluation rollouts during training for EMD, SAC, TD3, and TQC. We use five different random seeds for training, the performance of all curves is the average of 10 evaluations every 5000 steps. The shaded area of the curve corresponds to the standard deviation, and all curves are smoothed.

The results show that in most environments, EMD performs comparably to TQC, and its performance does not decrease as a result of reducing the number of networks and scaling down the model. However, in Ant environment, sudden turns by the agent can result in extremely low abnormal rewards. This unexpected event greatly affects the value of the scale parameter in the output of the critic network. And this phenomenon persists throughout the entire training process, leading to lower results compared to TQC. In Humanoid environment, the difficulty of the environment causes the agent to frequently experience sudden "deaths" during early training, which also has a significant impact on the value of the scale parameter in the output of the critic networks. However, with sufficient training experiences, the agent can improve stably without frequent deaths, resulting in excellent results after adequate training. Compared to SAC and TD3, which have network architectures closer to ours, our EMD algorithm outperforms them in all environments.

## 5.3 ABLATION STUDY

In this section, we will demonstrate the crashing phenomenon that occurs in some seeds of Walker2d environment when using one single critic with EMD and provide the appropriate analysis. We also examine how sensitive EMD is to hyperparameters: the number of atoms selected from the distribution and the number of atoms discarded.

**Crashing Phenomenon.** In previous experiments, Fig. 1(e) shows that such one critic network that outputs the minimum values of the location and scale parameters, respectively, will crash from time to time. In this section, we discuss it further. First, we present the results of using our dual-critic EMD as well as the results of using one single-critic EMD with three seeds (one of which experienced a crashing phenomenon), as shown in Fig. 3.

From Figure 3(a), it is evident that the average score of the single-network EMD is significantly lower than that of the double-critic EMD. Nevertheless, upon examining the results of each individual seed, it becomes apparent that the single-network EMD possesses comparable potential to the double-critic EMD, as long as there are no crashes.

Additionally, as shown in Fig. 3, by comparing the crashed seeds and the normal seeds where no crash occurred, we can identify the cause of the crash: consecutive anomalous values that cause the scale parameter to increase. By observing the scale parameter values of the normal seeds, we can see that a single occasional crash value does not necessarily lead to a complete crash of the process. The continuous anomalous values that appear in the experiment represented by the crashed seeds can make the model irretrievable, with the scale parameter remaining at the maximum value $(e^4)$ defined by us, losing the representational capability of the model. Using double-critic architecture can alleviate the crash to some extent, as the probability of both networks experiencing consecutive crashes becomes very low. Since having more critic networks does not seem to significantly improve the performance of the model and can slow down the training speed, we do not consider using more critic networks.

**Hyperparameter Settings.** We test the effect of different parameters on the experimental results on Halfcheetah environ-

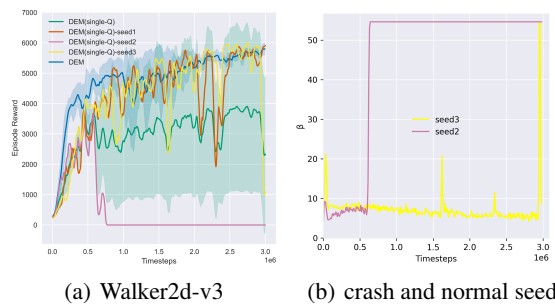

(a) Walker2d-v3  (b) crash and normal seed

Figure 3: Crash in single-Q.

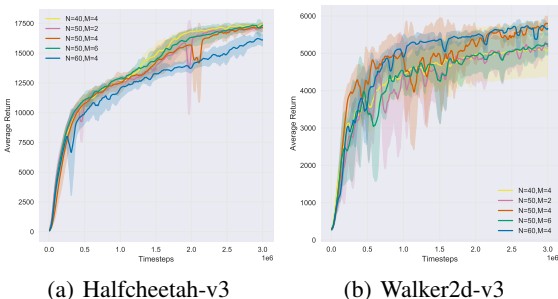

(a) Halfcheetah-v3  (b) Walker2d-v3

Figure 4: Hyperparameter settings.

ment, which is subject to estimation bias is distinctive, and Walker2d—an environment with the largest sizes of action and observation spaces. The main hyperparameters involved in our experiments are the number of atoms selected from the distribution and the number of atoms discarded. We test the experimental results for some reasonable hyperparameter settings within a certain range (the Extreme Value Theorem requires that the number of atoms thrown away be a small range of the total number of atoms), as shown in Fig. 4.

As shown in Fig. 4, our method is hyperparameter insensitive. Similar results are achieved at most of these reasonable hyperparameter settings. Therefore, we use the same hyperparameter settings in previous performance tests as in TQC to make our comparison more convincing. Strictly speaking, without affecting the performance, the smaller the number of atoms selected from the distribution, the shorter the time required to train the same number of steps by our method.

## 6 CONCLUSION

In this paper, we reinterpret the rationality of truncation in DRL using Extreme Value Theorem and derive the property that the distribution of truncated DRL gradually approaches the minimum value distribution. After that, we clarify the definition of extreme minimum distribution and some of its properties similar to the Gumbel distribution and give proof to clarify the rationality of our subsequent method. Based on the theoretical results, we remodel the critic network in the actor-critic method by using a critic that outputs the location and scale parameters of the extreme minimum distribution instead of the traditional critic in SAC and update it using a similar method to QDRL. Finally, we evaluate DEM in continuous control environments to demonstrate its advantage.

The updating method of the extreme minimum distribution determines the performance of the algorithm. We achieves good results using the 1-Wasserstein metric. Some other measurement methods, such as KL divergence, are limited in our approach due to computational difficulties. In the future we want to try more methods. In addition, we believe that the Extreme Value Theorem can play a greater role in RL, it deserves more discussion in RL community.

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

## A PROOF

Due to the symmetrical structure of the extreme minimum distribution and Gumbel distribution, they have very symmetrical properties. But they are rarely mentioned in previous papers. Here, we provide a simple proof using the properties related to Gumbel distribution. The relevant properties of these extreme minimum distributions are not our work.

### A.1 PROOF OF THE THEOREM IN THE MAIN TEXT

**Theorem 3** (Extreme Minimum Distribution) (Fisher & Tippett, 1928). For i.i.d. random variables $X_1, \cdots, X_n \sim f_X$, with left exponential tails, $\lim_{n \to \infty} \min_i (X_i)$ follows the extreme minimum distribution $\mathcal{F}(\mu, \beta)$, which has PDF $f(x) = exp(z - e^z)/\beta$ and CDF $F(x) = 1 - \exp(-e^z)$, where $z = (x - \mu)/\beta$ with location parameter $\mu$ and scale parameter $\beta$.

*Proof: For n sets of i.i.d. random variables, $X_1, \cdots, X_n \sim f_X$, the location parameter of the distribution of $\lim_{n \to \infty} \min_i (X_i)$ is $\mu$. We consider the opposite of these sets of random variables $-X_1, \cdots, -X_n \sim f_{-X}$, according to the definition of the Gumbel distribution, we can get the distribution of the maximum value:*

$$\lim_{n \to \infty} \max_i (-X_i) \sim \mathcal{G}(-\mu, \beta) \tag{13}$$

*where $-\mu$ is location parameter and $\beta$ is scale paramete of the distribution $\mathcal{G}$, and the CDF is $G(-x) = \exp\left(-e^{-\frac{-x-(-\mu)}{\beta}}\right)$. Since $\lim_{n \to \infty} \max_i (-X_i) = -\lim_{n \to \infty} \min_i (X_i)$, the CDF of the extreme minimum distribution can be writted by:*

$$
\begin{aligned}
F(x) &= P(X \le x) \\
&= 1 - P(X > x) \\
&= 1 - P(-X < -x) \\
&= 1 - G(-x) \\
&= 1 - \exp\left(-e^{\frac{x-\mu}{\beta}}\right).
\end{aligned}
\tag{14}
$$

*Taking the derivative of Eq. 14 yields the PDF of the distribution:*

$$f(x) = \frac{1}{\beta} \exp\left(\frac{x - \mu}{\beta} - e^{\frac{x-\mu}{\beta}}\right). \tag{15}$$

**Theorem 4** (Min-stable) (Mood, 1963). $\mathcal{F}$ is min-stable, i.e. if $X_i \sim \mathcal{F}$, then $\min_i (X_i) \sim \mathcal{F}$ holds.

*Proof: Consider $X_i \in \mathcal{F}(\mu, \beta)$, $\min_i (X_i) = -\max_i (-X_i) \sim -\mathcal{G}(-\mu, \beta)$. Same as Eq. 14, we can get $\min_i (X_i) \sim \mathcal{F}$.*

**Theorem 5** (Hazan & Jaakkola, 2012). For i.i.d. $\epsilon_i \sim \mathcal{F}(0, \beta)$ added to a set $\{x_1, \ldots, x_n\} \in \mathbb{R}$, $\min_i (x_i + \epsilon_i) \sim \mathcal{F}(-\beta \log \sum_i \exp(-x_i/\beta), \beta)$.

Before giving the proof of this theorem, we give Lemma 1 to help the proof of Theorem 5.

**Lemma 1** (Gumbel-Max Trick) (Hazan & Jaakkola, 2012). For i.i.d. $\epsilon_i \sim \mathcal{G}(0, \beta)$ added to a set $\{x_1, \cdots, x_n\} \in \mathbb{R}$, $\max_i (x_i + \epsilon_i) \sim \mathcal{G}(\beta \log \sum_i \exp(x_i/\beta), \beta)$

We then prove Theorem 5.

*Proof:*     *For i.i.d.  $\epsilon_i \sim \mathcal{F}(0, \beta)$  and a set  $\{x_1, \ldots, x_n\} \in \mathbb{R}$, by comparing the expressions of  $\mathcal{F}(0, \beta)$  and  $\mathcal{G}(0, \beta)$, there are  $-\epsilon_i \sim \mathcal{G}(0, \beta)$, then we can get  $\min_i (x_i + \epsilon_i) = -\max_i (-x_i + (-\epsilon_i)) \sim -\mathcal{G}(\beta \log \sum_i \exp(-x_i/\beta), \beta)$, Same as Eq. 14, according to Gumbel-Max Trick, we have  $\min_i (x_i + \epsilon_i) \sim \mathcal{F}\left(-\beta \log \sum_i \exp(-x_i/\beta), \beta\right)$.*

With these theories above, we consider Theorem 2.

We use $F_0(x)$ to represent the initial distribution, $\mathcal{G}_i(x) (i > 0)$ to represent the distribution of extreme atoms truncated at the $i$-th iteration, and $F_i(x) (i > 0)$ to represent the remaining distribution after the $i$-th truncation.

**Theorem 2** (Distribution of Truncate DRL). For a distribution of any shape, there exists a truncated number $N$ of truncations to the right tail of times, $\forall n > N$, such that the remaining distribution after $n$ truncations follows the extreme minimum distribution. In RL, truncating the maximum output of the critic network will cause the remaining portion to follow the extreme minimum distribution.

Let's understand this from a simple example. A piece of string, we cut off the 1/10th of the very tail end each time, after enough times of truncation operation, the remaining string must be a small part of the very front end of the original string.

Then we prove the theorem mathematically.

*Proof:*     *Let us consider an arbitrary distribution function, whose cumulative density function we denote by $F(x)$. When truncating, we mark the cumulative density function of the distribution before truncation as $F_{before}$, and the cumulative density function of the distribution after truncation as $F_{after}$. By the property of truncated distribution, we can write the distribution of the tail atoms after truncating the $\tau$ as:*

$$F_{after}(x) = F_{before}(x) / F_{before}(\tau) \tag{16}$$

*Therefore, we perform the same operation during each truncation process (assuming that the proportion of truncation is the same each time for ease of computation, truncature the parts of $(\tau_{tru}, 1]$, which has no effect on the final result). After any number of truncations, the remaining distribution $F_n(x)$ can be expressed as:*

$$
\begin{aligned}
F_n(x) &= F_{n-1}(x) / F_{n-1}(\tau_{del}) \\
&= F_{n-2}(x) / (F_{n-1}(\tau_{del}) F_{n-2}(\tau_{del})) \\
&\vdots \\
&= F_0(x) / (F_{n-1}(\tau_{del}) F_{n-2}(\tau_{del}) \cdots F_0(\tau_{del}))
\end{aligned}
\tag{17}
$$

*For the CDF at these quantiles, we have $0 < F_{n-1}(\tau_{del}) \leq \cdots \leq F_0(\tau_{del}) < 1$. We use $K(n) = F_{n-1}(\tau_{del}) F_{n-2}(\tau_{del}) \cdots F_0(\tau_{del})$, we have $\lim_{n \to +\infty} K(n) = 0^+$. Now we use $\tau$ to represent the truncation point, and the distribution of the left part of $\tau$ after truncation can be considered as a minimum distribution. For $\forall \tau$, we have $F_0(\tau) > 0$, there exists an integer $N$ for all $n > N$, has $K(n) < F_0(\tau)$. So as the distribution is continuously truncated, there will always be $K(n) < F_0(\tau)$. The distribution of the remaining portion after truncation can be considered as the extreme minimum distribution. One point to note is that $K(n)$ is an exponential function of $n$, so in practical applications, we do not need many truncation steps to consider the remaining distribution as the minimum distribution, especially compared to RL algorithms that require millions of training steps.*

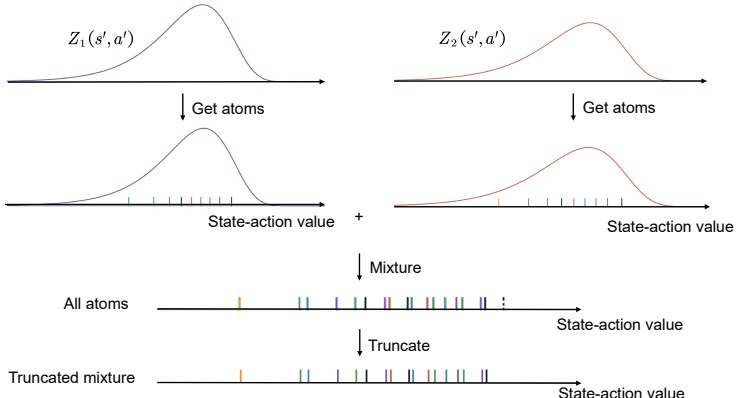

Figure 5: Selection of atoms for the temporal difference target distribution.

## A.2 THE SPECIFIC DERIVATION OF EQUATION 11

For a distribution, we usually use $\int x f(x) dx$ to calculate its distribution. We can get the value of expected discounted reward $Q$ by:

$$
\begin{aligned}
Q_i(s,a) &= \int x f(x; \mu_i, \beta_i) dx \\
&= \int x \left( \frac{1}{\beta_i} \exp \left( \frac{x - \mu_i}{\beta_i} - e^{\frac{x - \mu_i}{\beta_i}} \right) \right) dx \\
&= \int (\beta_i u + \mu_i) \left( \frac{1}{\beta_i} \exp (u - e^u) \right) \beta_i du \\
&= \mu_i \int \exp (u - e^u) \, du + \beta_i \int u \left( \exp (u - e^u) \right) du \\
&= \mu_i - \lambda \beta_i, i = 1, 2.
\end{aligned} \tag{18}
$$

## B ADDITIONS TO THE ALGORITHM

Here we add a flowchart to the process described in 4.2.1, and we give the specific flow of the algorithm in this section.

## B.1 FLOWCHART FOR OBTAINING TARGET DISTRIBUTION

The specific process has been written in the main text. We only briefly describe here the process shown in Fig. 5.

For the extreme minimum distribution output of two target networks, we use Equation 7 to sample their values at each quantile and then mix them together. Remove the largest small part of the mixed values and update the two critic networks with the remaining atoms.

It is worth noting that we do not re-represent the truncated atoms as distributions because we do not use the truncated specific distributions in the updating process of the network, only the quantile of the distribution.

## B.2 THE ALGORITHM OF EMD

---

**Algorithm 1** EMD

---

1: Initialize critic networks $Q_{\theta_1}$, $Q_{\theta_2}$, and actor network $\pi_\phi$ with random parameters $\theta_1, \theta_2, \phi$
2: Initialize target networks $\theta'_1 \leftarrow \theta_1, \theta'_2 \leftarrow \theta_2, \phi' \leftarrow \phi$
3: Initialize the replay buffer $\mathcal{D} = \varnothing, \eta = 0.005$.
4: **for** each iteration **do**
5:     **for** each environment step **do**
6:         $a_t \sim \pi_\phi(a_t|s_t)$
7:         $s_{t+1} \sim p(s_{t+1}|s_t, a_t)$
8:         $\mathcal{D} \leftarrow \mathcal{D} \cup \{(s_t, a_t, r(s_t, a_t), s_{t+1})\}$
9:     **end for**
10:     **for** each gradient step **do**
11:         sample a batch from the replay $\mathcal{D}$
12:         Train $Q_{\theta_1}$ and $Q_{\theta_2}$ using $\mathcal{L}(s, a; \theta_i)$ from Eq. 10
13:         Train $\pi_\phi$ using $J_\pi(\phi)$ from Eq. 12
14:         $\theta'_i \leftarrow \eta\theta_i + (1-\eta)\theta'_i, i = 1, 2$
15:         $\phi' \leftarrow \eta\phi + (1-\eta)\phi'$
16:     **end for**
17: **end for**

---

Table 1: Hyperparameters values.

| Hyper-parameter | Ours | TQC | SAC | TD3 |
|---|---|---|---|---|
| Actor Learning Rate | $3 \times 10^{-4}$ | $3 \times 10^{-4}$ | $3 \times 10^{-4}$ | $1 \times 10^{-3}$ |
| Optimizer | Adam | Adam | Adam | Adam |
| Target Update Rate $\eta$ | 0.005 | 0.005 | 0.005 | 0.005 |
| Batch Size | 512 | 512 | 512 | 512 |
| Iterations per time step | 1 | 1 | 1 | 1 |
| Discount Factor | 0.99 | 0.99 | 0.99 | 0.99 |
| Replay Buffer Size | $1 \times 10^6$ | $1 \times 10^6$ | $1 \times 10^6$ | $1 \times 10^6$ |
| Number of Critics | 2 | 5 | 2 | 2 |
| Nonlinearity | 256 | 512 | 256 | 256 |
| Number of Hidden Layers in Critic Networks | ReLU | ReLU | ReLU | ReLU |
| Entropy Temperature Coefficient $\alpha$ | 0.2 | 0.2 | 0.2 | - |
| Number of Single Network Atoms $N$ | 25 | 25 | - | - |
| Huber Loss Parameter $\kappa$ | 1 | 1 | - | - |

## C ADDITIONS TO THE EXPERIMENT

### C.1 DETAILED PARAMETERIZATION OF THE EXPERIMENT

During the experiments, the parameters of our comparison methods are set according to the parameters in the corresponding papers. The specific parameter settings are shown in Table 1.

Additionally, we need to clarify one point about our critic network. The final location parameter and scale parameter obtained by our critic network are output from network layers that are not directly connected to each other to ensure their unique characteristics.

It is also important to note that we clip the scale parameters of the output of the critic network to ensure that the shape of the resulting distributions remained within a certain reasonable range during the experiment.

Specifically, the architecture of our critic networks is shown in Figure 6.

In the Ant environment, the clipping range for our scale parameter setting is $[-20, 3]$ to make sure that the distribution does not crash although this may reduce the experimental results. In other environment, the clipping range for scale parameter setting is $[-20, 4]$.

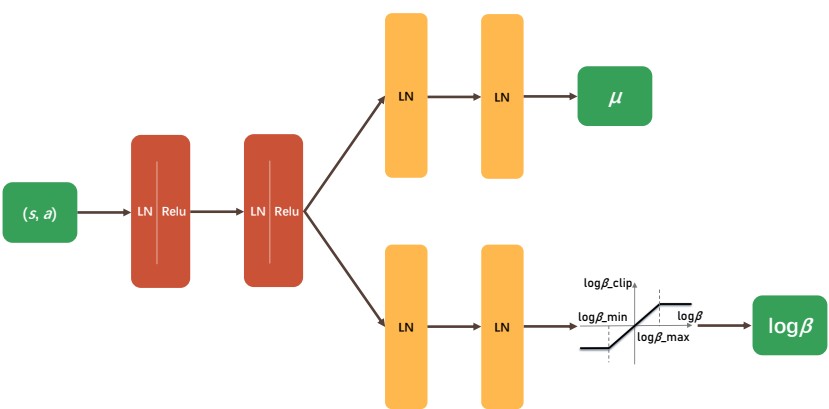

Figure 6: The architecture of critic networks.

## C.2 Comparison of experimental time with other methods

Figure 7 shows the run time taken for the first 100 thousand training steps for EMD, SAC, TD3, and TQC training on an Intel i7-11700K CPU. We use the average of the 5 environments as the final result and the black lines represent the time in each environment.

Our method uses fewer critic networks, and fewer critic network outputs relative to TQC, and so has a shorter training time. Compared to SAC, our method uses a more complex critic network and a more cumbersome loss function, so it has a longer training time. However, we believe that the loss function of our architecture can be optimized in the future in terms of method and code, and the training time can be further reduced. It should be noted that our time experiments are conducted on a single CPU, which needed to exclude the influence of other processes. Except for the time experiment, all other experiments are conducted on an Inter(R) Xeon(R) Platinum 8350C CPU and a NVIDIA GeForce RTX 3090 GPU.

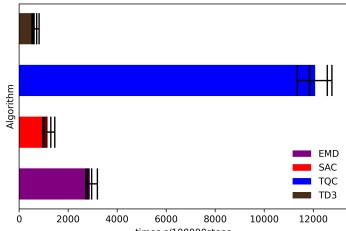

Figure 7: Time spent.

## D For a discussion of three types of Extreme Value Theorem

In this section, we provide a detailed introduction to the applicability of three types of Extreme Value Theorem in RL. Type I, II, III extreme value distribution also known as Gumbel distribution, Fréchet distribution, and Weibull distribution, they can be seen as tail distributions of exponential distribution, power-law distribution, and distributions with supremum/infimum. In the continuous domain task of RL, considering an infinite Markov process, we can express a possible discount reward of current $(s_t, a_t)$ as:

$$
\begin{aligned}
z\left(s_t, a_t\right) &= r\left(s_t, a_t\right) + \gamma p_{s_t, a_t} \pi_{s_{t+1}}\left[z\left(s_{t+1}, a_{t+1}\right)\right] \\
&= r\left(s_t, a_t\right) + \gamma p_{s_t, a_t} \pi_{s_{t+1}}\left[r\left(s_{t+1}, a_{t+1}\right) + \gamma p_{s_{t+1}, a_{t+1}} \pi_{s_{t+2}}\left[z\left(s_{t+2}, a_{t+2}\right)\right]\right] \quad (19) \\
&= \cdots
\end{aligned}
$$

The probability of its occurrence is $q = p_{s_t, a_t} \pi_{s_{t+1}} p_{s_{t+1}, a_{t+1}} \pi_{s_{t+2}} \cdots$. In existing research, it is generally believed that $z\left(s_t, a_t\right)$ has no clear supremum/infimum, so we do not consider Weibull distribution. But we cannot provide a clear definition of the distribution shape of the $z\left(s_t, a_t\right)$, so we need to consider both Gumbel distribution, and Fréchet distribution. In the main text, we

discussed a method of using Gumbel distribution to model the output of critic network. Here, we consider Fréchet distribution.

The CDF of Fréchet distribution is

$$Pr\left(X \leq x\right) = \begin{cases} e^{-\left(\frac{x-\mu}{\beta}\right)^{-\iota}} & x > \mu \\ 0, & x \leq \mu \end{cases} \tag{20}$$

where $\iota$ is shape parameter. Like the Gumbel distribution, this distribution is used to represent the right tailed distribution. We hereby provide its corresponding form representing the left tailed distribution:

$$Pr\left(X \leq x\right) = \begin{cases} 1 - e^{-\left(\frac{-x+\mu}{\beta}\right)^{-\iota}} & x < \mu \\ 1, & x \geq \mu \end{cases} \tag{21}$$

Note that Fréchet distribution has an additional shape parameter $\iota$ compared to the Gunbel distribution, so our critic network outputs an additional shape parameter on top of the original output. So, when we apply the Fréchet distribution, we need to replace Eq. 7 with

$$Z_{tar}\left(s, a\right) = F^{-1}\left(\tau; \mu_{tar}, \beta_{tar}, \iota_{tar}\right) = \mu_{tar} - \beta_{tar}\left(-\ln\left(1 - \tau\right)\right)^{-\frac{1}{\iota}}, \tag{22}$$

In addition, the calculation of the mean of Fréchet distribution is relatively complex. We use the average of the results at all sampling quantiles of the critic network to update the policy network.

The rest of the settings are the same as the main text.

Fig. 8 shows the experimental results of using Fréchet distribution in the mujuco environment. Among them, EMD−I represents using the type I extreme minimum distribution to model the critic network, and EMD−II represents using the type II extreme minimum distribution. The experimental results indicate that modeling critic networks using Fréchet distribution is feasible. However, compared to the Gumbel distribution, the results in the vast majority of environments are worse. Therefore, we believe that using the form of the extreme minimum distribution of type I extreme value distribution is more suitable for the distribution of the output results of critic networks.

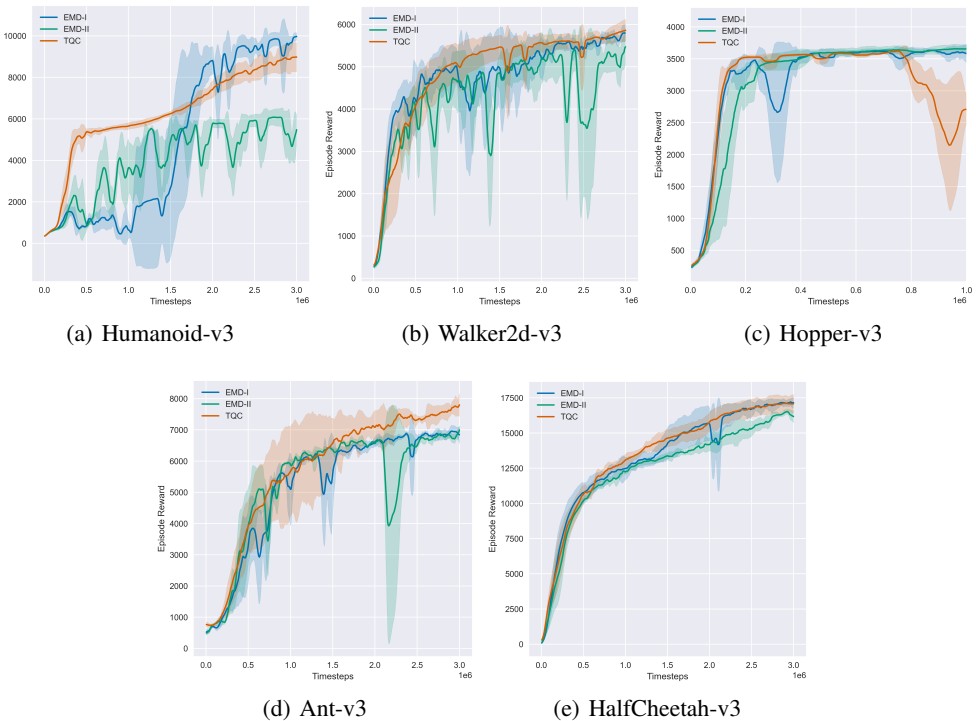

Figure 8: Training curves on continuous control benchmarks.

For critic networks using type $II$ extreme minimum distribution, the clipping range for scale parameters is $(e^0, e^4)$, and the reference range for shape parameters is $(e^0, e^3)$. The other settings are the same as using the critic networks with type $I$ extreme minimum distribution.

# E CONVERGENCE OF DISTRIBUTED BELLMAN OPERATORS UNDER EXTREME VALUE TRUNCATION

For Theorem 5, we consider "min" as a truncation operation, $\epsilon$ as the offset noise accompanying the single step reward $r$, and $r$ itself as the overall offset of the distribution. We get the truncated Bellman operation:

$$\mathcal{T}^t Z(s, a) = truncate\left(r + \gamma E_{s' \sim p, a' \sim \pi} Z(s', a')\right). \tag{23}$$

Note that $\epsilon$ is included in $r$.

The EMD algorithm optimizes by minimizing the 1-Wasserstein distanc between $Z_i(s, a), i = 1, 2$ and $Y(s, a)$ when updating critic networks. Formally, it can be written as $\int_0^1 \left|Z^{-1}(s, a; \tau) - Y^{-1}(s, a; \tau)\right| d\tau$. Considering each possible $\tau \in [0, 1]$, we can find that the difference between the atoms at each quantile in our method and SAC is that the introduction of truncation makes $z_\tau^{after\_truncation}(s, a) \leq z_\tau^{before\_truncation}(s, a)$. We consider the soft Bellman equation with truncation:

$$z_{\pi_{old}}^{after\_truncation}(s_t, a_t)$$
$$\leq r(s_t, a_t) + \gamma \mathbb{E}_{s_{t+1} \sim p}\left[\mathbb{E}_{a_{t+1} \sim \pi_{new}}\left[z_{\pi_{old}}^{after\_truncation}(s_{t+1}, a_{t+1}) - \log \pi_{new}(a_{t+1}|s_{t+1})\right]\right]$$
$$\leq r(s_t, a_t) + \gamma \mathbb{E}_{s_{t+1} \sim p}\left[\mathbb{E}_{a_{t+1} \sim \pi_{new}}\left[z_{\pi_{old}}^{before\_truncation}(s_{t+1}, a_{t+1}) - \log \pi_{new}(a_{t+1}|s_{t+1})\right]\right]$$
$$\tag{24}$$

In this way, we can conclude that this difference does not affect the convergence of the soft Bellman equation. We will not explain the convergence of the soft Bellman equation again here because the proof in SAC (Haarnoja et al., 2018) is good enough. Now, we can assume that the atoms at each quantile are convergent. However, even if the values at the quantile are considered convergent alone, we cannot assume that the entire distribution is convergent. We cannot provide a complete explanation of the convergence of Distributed Bellman Operators under Extreme Value Truncation.

