# OpenReview forum: "Reinforcement Learning with Extreme Minimum Distribution"
_ICLR.cc/2024/Conference — ICLR 2024 Conference Withdrawn Submission_

### Official Review · Reviewer_eazc · 2023-10-29

**Soundness:** 1 poor
**Presentation:** 2 fair
**Contribution:** 1 poor
**Rating:** 1
**Confidence:** 4

**Summary:**

The paper incorporates some results in Extreme Mean Theory into distributional RL, especially in the continuous control setting. As the large atoms used in distributional RL may lead to unstable optimization, the authors proposed to discard them. It seems that the remaining part follows an extreme minimum distribution, allowing to directly output the scale and location parameter for the asymptotical Gumbel distribution. A heuristic algorithm is designed in the actor-critic framework by simply discarding the max atoms and shows a similar performance compared with the existing baselines.

**Strengths:**

* The paper is easy to understand as the method is straightforward.

**Weaknesses:**

* **The novelty and technical contribution are trivial and very limited**. Since TQC has already considered the truncation technique to stabilize the RL training process, it is very incremental to use an already known Extreme value theorem to consider this problem again without any technical contribution. Besides, the authors claimed the advantage of the proposed algorithm over TQC is less computational cost, which is not important as both the two algorithms already largely increase the computation burden compared with other distributional RL algorithms. For example, EMD uses critic networks to output scale and location parameters, leading to unstable concerns compared with other distributional RL baselines with only one critic. More importantly, the performance of EMD is very similar to TQC.

* **The methodology is questionable and does not have justifications**. The authors are trying to borrow some existing statistical techniques or conclusions into RL problems without investigating whether those results are valid under real problems. Generally speaking, some existing results in statistics typically rely on distribution assumptions, which are normally not applicable to RL problems without rigorous justifications. In particular, Extreme value theory includes a few types and the paper here only considers the iid case with exponential family. Note that in online RL training, it is not clear whether the resulting distribution is exponential or not and I am afraid no in general based on my knowledge. Therefore, the proposed method relies on a very strong distribution assumption, which is typically not valid in complex environments. Also, the Gumbel distribution is an asymptotical result, and in practice, we are more likely to care about the non-asymptotical scenario. I thus doubt the rationale behind the proposed algorithm let alone the fact that the empirical improvement is very limited.

* **Necessary theoretical parts are missing.** To begin with, I double that the theorems presented in this paper are very likely directly based on the existing results in the extreme value theorem. Without providing the reference, the paper would suffer from the academic integrity issue. Apart from that, it is not clear whether the distributional Bellman operator under the extreme value truncation is convergence or not, which should be rigorously discussed. In addition, Theorem 2 is provided in a very non-mathematical way, which is less rigorous and not convincing to me.

* **Missing literature.** Truncating the extreme value is also highly linked with risk-sensitive RL, but the relevant literature is missing. It is not clear whether truncating the extreme value is helpful or not as it is a trade-off between exploration and stability. However, the paper fails to justify this trade-off rigorously.

* **Experiments.** Experiments are very restricted in continuous control cases, and improvement is not significant in Figure 2. Since the overestimation issue is more commonly used in value-based RL, experiments on Atari games are necessary and the current experimental results are weak.

* **Writing**. The writing should be substantially improved and the choice of some words is causal. Some paragraphs like continuous control algorithms in the literature are well-known. The most of parts in Section 4.2 are trivial and similar to existing works.

**Questions:**

Please refer to the Weakness part.

---

> ### Author Response · Authors · 2023-11-20
> **Official Comment by Authors (Part 1)**
>
> Firstly, we would like to thank you very much for your reply and suggestions, which are very important for improving our current and future work. In addition, we apologize for our poor writing and we will focus more on that.
>
> [W1.1: Technical contribution] Simply using the extreme value theorem to consider this approach is indeed incremental, but our goal is to introduce the following substitution of quantile networks with critic networks with extreme minimum distribution, which has not appeared in our known RL methods.
>
> [W1.2: Computational cost and the empirical evidence] We believe that it is not appropriate to discuss calculation costs without considering performance, and comprehensive consideration should be given. Compared to other distributed RL methods, our method and TQC mostly have better results. To our knowledge, TQC is one of the most competitive methods among existing distributed methods in the MuJoCo environment under the Gym framework.
>
> [W1.3: Instability] We acknowledge that the presence of scale parameters makes the network unstable. In this paper, we discussed this issue and greatly reduced the impact of this instability on the method by using parameter truncation and the use of double critic networks.
>
> [W2.1: On Extreme Value Theory] Thank you very much for pointing out. At the beginning, we think that the other two extreme value distributions have a strict probability interval of 0, which is generally considered non-existent in the distribution RL, so we do not consider other distributions. After reading your comments, we have reconsidered this issue and believe that it does require some experimentation or derivation for discussion. We discuss this in Appendix D and explain it in the main text. In addition, we need to clarify that consider the tails of the distribution as Gumbel distribution does not need the results to satisfy an exponential distribution, only the results to have an exponential tail.
>
> For type-III extreme value distribution, it requires the distribution to have supremum/infimum. While in DRL, it is generally believed that the probability at the tail of the distribution approaches 0 infinitely but does not equal 0. Therefore, we do not consider type-III extreme value distribution.
>
> For other two types of extreme value distributions, we have supplemented the performance experiments of type-I extreme value distribution on the basis of the original use of type-I extreme value distribution, and presented the experimental results in Appendix D.
> Furthermore, we acknowledge that we cannot specify which shape the result distribution of an unguided critic network should be, as this is very difficult.
>
> Our Theorem 2 is for all distributions, and now, approaching it, we consider the full range of theoretically feasible extreme value distributions.
>
> [W2.2: Asymptotical] We acknowledge that the theory of our method is asymptotic, but for so many training steps of existing RL methods, the output of the critic network of these truncated methods will strictly become extreme minimum distribution within an extremely short training step, which is negligible for high RL training steps.
>
> [W2.3: The empirical improvement] Our views on improvement: We redesigned the critic network based on the theory that the truncation update method will make the output of the critic network conform to the extreme minimum distribution. We used a reasonable update method and achieved very competitive experimental results compared to the current RL method. I think there is a certain contribution.
>
> [W3.1: Reference] Thank you for your correction. As shown by the distribution function of the extreme minimum distribution, it is a symmetric form of the Gumbel distribution about a certain x-value, all of which have very similar properties. However, in the literature I have read, they only use maximum values as representatives to derive several types of extreme maximum distribution (including Gumbel distribution), and say the extreme minimum distribution has similar properties, without providing specific expressions and properties of the extreme minimum distribution. Therefore, we use the relevant content of existing Gumbel distribution to provide a detailed representation and derivation process of the corresponding properties of the minimum distribution. Because this is necessary for our method (the distribution form is used for network output, and the properties are used to ensure that the distribution form does not change during the RL update process).
>
> But we do not provide references, it was our mistake. Now, we have provided the literature and we hope that everyone can supervise us. If there is any dispute, we will immediately revise our paper to ensure that there is no academic misconduct.

---

> ### Author Response · Authors · 2023-11-20
> **Official Comment by Authors (Part 2)**
>
> [W3.2: Astringency] We demonstrate in Appendix E that our method converges at any quantile. However, convergence for any quantile does not necessarily mean convergence for the entire distribution. We cannot demonstrate the convergence of the entire method, which is the weak point of our work. We will conduct more research in the future.
>
> [W3.3: Theorem 2] We rephrase Theorem 2 to make it more rigorous. In addition, we re-prove the theorem (although we also used the relevant properties of truncated CDF), making it more mathematical and persuasive.
>
> [W4: Risk sensitive perspective] From a risk sensitive perspective, our approach is conservative, somewhere between risk-averse and risk-neutral. Because EMD is equivalent to discarding the highest predicted part of the results distribution of critic network. I think our contribution to risk sensitivity is very weak, and what we can say is already excellent enough in previous paper[1] with citations, so we don't need to discuss it separately.
>
> [1] Jihwan Oh, Joonkee Kim, Se-Young Yun. Risk Perspective Exploration in Distributional Reinforcement Learning. arXiv preprint arXiv:2206.14170 (2022).
>
> [W5: Experiments] Our method is to replace the common practice of discarding larger values and retaining smaller values in solving overestimation problems (Such as TD3, EDAC in offline RL, etc.) using a single or fewer networks. Get better or equivalent results with less overhead. In value-based RL, the more common method to solve overestimation is Double DQN, which does not adopt the method of selecting the smaller value from multiple critic evaluation results and rounding off the larger value. There is a difference from our design philosophy, so no relevant experiments are conducted.
>
> For evaluations of restricted experiments result, as we mentioned earlier, TQC has the most competitive results under continuous control, and our method can achieve similar or better experimental results with less overhead in most environments.
>
> [W6: Writing] I apologize for my poor writing. We will pay more attention to this. For section 4.2, we acknowledge that the writing in this section is not good enough. However, in this section, we introduce the specific implementation process of our method, which I believe is necessary.
>
> We have revised our paper:
> 1) We rephrase Theorem 2 to make it more rigorous. In addition, we have re proved the theorem (although we also used the relevant properties of truncated CDF), making it more mathematical and persuasive.
> 2) Added Appendix D. Discuss the other two forms of minimum distribution and demonstrate the performance experimental results of feasible methods. And add relevant descriptions in the main text (under Theorem 2).
> 3) Added Appendix E. Simply explain that our method converges at any quantile.
>
> We sincerely hope that our response has addressed your questions and will contribute to improving the final score of the paper. If you have any further questions or suggestions, please don't hesitate to let us know. We would be more than happy to provide clarification and answer any additional inquiries you may have.
>
> Thank you again for your feedback, they are very helpful to me.

---

### Official Review · Reviewer_oowZ · 2023-11-01

**Soundness:** 2 fair
**Presentation:** 2 fair
**Contribution:** 2 fair
**Rating:** 3
**Confidence:** 4

**Summary:**

The paper proposes a solution to tackle the tail behavior of the return distribution. Specifically, the paper truncates the tail atoms and derives the truncated distribution in the asymptotic setting using the Extreme Value Theorem. The critic network can be devised accordingly and the paper provides empirical verification of the proposed method.

**Strengths:**

N/A

**Weaknesses:**

### The theoretical part of the work is not rigorous enough to be ready for a conference paper.

1. Theorem 2 is not rigorous. What is an "iterative process"? What is the mathematical definition for "removing extreme maximal atoms"?  The Proof of Theorem 2 in the appendix is also far-fetched and hard to understand. Why does $\alpha^n \to 0+$ imply "we can maintain $F_n(x)$ represents the smallest portion of the values"?

2. Theorem 3: what is "exponential head"?

### The significance of the theoretical part is not enough.

- Theorems 3~5 are just simple manipulations of the distribution.

### The empirical evidence is not strong enough.

- In Figure 2, TQC mostly performs better or equal to the performance of the proposed EMD.

### The clarity of the paper needs to be improved.

**Questions:**

See above

---

> ### Author Response · Authors · 2023-11-20
>
> Thank you very much for your evaluation and suggestions.
>
> [W1.1: Theorem 2] Thank you very much for your advice! We first apologize for our lack of strict vocabulary. We rephrase Theorem 2 to make it more rigorous. In addition, we re-proved the theorem (although we also use the relevant properties of truncated CDF), making it more mathematical and persuasive. Your suggestions for new descriptions and theorem-proving methods are welcome.
>
> [W1.2: Theorem 3] Thank you for pointing out. This is our mistake. We originally intended to distinguish the left tail from the right tail of the distribution, but as a result, we made an incorrect statement. It has now been modified.
>
> [W2: Theoretical part] (Theorems 3~5 are just simple manipulations of the distribution) Our main theory is Theorem 2, and the operation of truncating the distribution will eventually approach a extreme minimum distribution and maintain it. Theorem 3-5 is just a simple explanation of the extreme minimum distribution (because the extreme value theory is already very complete and the minimum is only one form of it, but we have not found a paper that directly explains these properties of the minimum distribution, so we have made a simple explanation to facilitate our understanding and use). Our Theorem 2 can be applied to any distribution form (we initially considered methods such as TQC and EDAC in offline RL, where selecting the minimum value and discarding the larger values. We want to replace the ensemble network with an extreme minimum distribution network), and it is simple and easy to implement, which has certain significance for the development of the RL community.
>
> [W3: The empirical evidence] (In Figure 2, TQC mostly performs better or equal to the performance of the proposed EMD) TQC is one of the methods with the highest scores within the MuJoCo environment under the Gym framework, and has better results compared to many recent algorithms. Our method performs slightly better than TQC in terms of performance (two are better than TQC, two are equivalent, and one is lower). Compared to TQC, our method has fewer networks and fewer parameters for a single network, which results in EMD training time of nearly 1/4 of TQC under the same number of training steps (Fig.7). Therefore, overall, we believe that our method is more competitive than TQC.
>
> [W4: The clarity of the paper needs to be improved] We apologize for our poor writing and we will focus more on that.
>
> We have revised our paper:
> 1) We rephrase Theorem 2 to make it more rigorous. In addition, we have re proved the theorem (although we also used the relevant properties of truncated CDF), making it more mathematical and persuasive.
> 2) Added Appendix D. Discuss the other two forms of minimum distribution and demonstrate the performance experimental results of feasible methods. And add relevant descriptions in the main text (under Theorem 2).
> 3) Added Appendix E. Simply explain that our method converges at any quantile.
>
> We sincerely hope that our response has addressed your questions and will contribute to improving the final score of the paper. If you have any further questions or suggestions, please don't hesitate to let us know. We would be more than happy to provide clarification and answer any additional inquiries you may have.

---

### Official Review · Reviewer_7cHh · 2023-11-04

**Soundness:** 2 fair
**Presentation:** 1 poor
**Contribution:** 4 excellent
**Rating:** 3
**Confidence:** 5

**Summary:**

This paper propose a new return distribution estimation method for distribution reinforcement learning.
The authors state that the tail of return distribution can worsen learning return distribution.
By using the property of Gumbel distribution, this paper conjectures that the topmost atoms of the approximated reward distribution are anomalies with large estimation bias by the limit of function approximator, such as quantile-regression networks.
To overcome this estimation bias, the proposed method re-estimate the inverse CDF by using extreme minimum distribution.
In experiment, the authors modify value-based distributional RL algorithms into a soft actor-critic variant to evaluate in the contiuous control domain.

**Strengths:**

The main method is  a novel correction method for existing quantile regression DQN.
The existing correction method for return distribution is a simple truncation method, but the propsed method can provide a theory-based distribution correction method, which has been already used in RL research (extreme Q-learning).

**Weaknesses:**

My main concern is the lack of the supporting materials for the main claim is not enough.
Although the propsed method might fit for distribution RL, the main theorem is not an analysis for distributional Bellman equation but a proposition of vanilla probability distribution.
In addition, the proposed algorithm cannot outpeform existing distributional RL baselines, such as TQC which has the similar concept and more simple computation structure.

**Questions:**

Please refer weakness section.

---

> ### Author Response · Authors · 2023-11-20
>
> Thank you very much for your evaluation and suggestions.
>
> [S: Difference with Extreme Q learning] Extreme Q learning is the first method in RL to use extreme value theory, which is a very powerful work. But our method is different, mainly used for re-modeling the truncated distribution RL method (More, re-model any method that discards larger to retain smaller values).
>
> [W1: Theorem] The results of our Theorem 2 are universal and applicable to all continuous cases of truncating the distribution (in a broad sense, we believe that this theorem is also closely related to the selection of smaller values for multiple Q (s, a) values while discarding larger values, which is the original intention of our method and subsequent work), including the distribution RL in the continuous action space. In addition, we give the truncated Bellman equation and add some views on the convergence of the truncated Bellman distribution equation in Appendix E.
>
> [W2: The empirical evidence] TQC is one of the methods with the highest scores in the MuJoCo environment under the Gym framework, and has better results compared to many recent algorithms. Our method performs slightly better than TQC in terms of performance (two are better than TQC, two are equivalent, and one is lower). Compared to TQC, our method has fewer networks and fewer parameters for a single network, which results in EMD training time of nearly 1/4 of TQC under the same number of training steps (Fig.7). Therefore, overall, we believe that our method is more competitive than TQC.
>
> We have revised our paper:
> 1) We rephrase Theorem 2 to make it more rigorous. In addition, we have re proved the theorem (although we also used the relevant properties of truncated CDF), making it more mathematical and persuasive.
> 2) Added Appendix D. Discuss the other two forms of minimum distribution and demonstrate the performance experimental results of feasible methods. And add relevant descriptions in the main text (under Theorem 2).
> 3) Added Appendix E. Simply explain that our method converges at any quantile.
>
> We sincerely hope that our response has addressed your questions and will contribute to improving the final score of the paper. If you have any further questions or suggestions, please don't hesitate to let us know. We would be more than happy to provide clarification and answer any additional inquiries you may have.